# Consequences of *Lmna* Exon 4 Mutations in Myoblast Function

**DOI:** 10.3390/cells9051286

**Published:** 2020-05-21

**Authors:** Déborah Gómez-Domínguez, Carolina Epifano, Fernando de Miguel, Albert García Castaño, Borja Vilaplana-Martí, Alberto Martín, Sandra Amarilla-Quintana, Anne T Bertrand, Gisèle Bonne, Javier Ramón-Azcón, Miguel A Rodríguez-Milla, Ignacio Pérez de Castro

**Affiliations:** 1Instituto de Investigación de Enfermedades Raras, Instituto de Salud Carlos III, Ctra. Majadahonda-Pozuelo km2.2, E-28029 Madrid, Spain; d.gomez@isciii.es (D.G.-D.); ferdemiguelpe@gmail.com (F.d.M.); bvilaplana@isciii.es (B.V.-M.); almmartin@isciii.es (A.M.); rmilla@isciii.es (M.A.R.-M.); 2Fundación Andrés Marcio, niños contra la laminopatía, C/Núñez de Balboa, 11, E-28001 Madrid, Spain; cepifano@hotmail.com; 3Universidad Europea de Madrid, C/ Tajo, s/n, E-28670 Villaviciosa de Odón, Spain; 4Institute for Bioengineering of Catalonia (IBEC), C/Baldiri Reixac, 10-12, E-08028 Barcelona, Spain; agarciac@ibecbarcelona.eu (A.G.C.); jramon@ibecbarcelona.eu (J.R.-A.); 5Fundación de Investigación HM Hospitales, Plaza del Conde Valle Suchil, 2, E-28015 Madrid, Spain; amarillaquintana@gmail.com; 6UMRS 974, Center of Research in Myology, Institut de Myologie, Sorbonne Université, INSERM, 75013 Paris, France; a.bertrand@institut-myologie.org (A.T.B.); g.bonne@institut-myologie.org (G.B.); 7ICREA-Institució Catalana de Recerca i Estudis Avançats, 08010 Barcelona, Spain

**Keywords:** LMNA, laminopathy, CRISPR, nuclear envelope

## Abstract

Laminopathies are causally associated with mutations on the Lamin A/C gene (*LMNA*). To date, more than 400 mutations in *LMNA* have been reported in patients. These mutations are widely distributed throughout the entire gene and are associated with a wide range of phenotypes. Unfortunately, little is known about the mechanisms underlying the effect of the majority of these mutations. This is the case of more than 40 mutations that are located at exon 4. Using CRISPR/Cas9 technology, we generated a collection of *Lmna* exon 4 mutants in mouse C2C12 myoblasts. These cell models included different types of exon 4 deletions and the presence of R249W mutation, one of the human variants associated with a severe type of laminopathy, *LMNA*-associated congenital muscular dystrophy (L-CMD). We characterized these clones by measuring their nuclear circularity, myogenic differentiation capacity in 2D and 3D conditions, DNA damage, and levels of p-ERK and p-AKT (phosphorylated Mitogen-Activated Protein Kinase 1/3 and AKT serine/threonine kinase 1). Our results indicated that *Lmna* exon 4 mutants showed abnormal nuclear morphology. In addition, levels and/or subcellular localization of different members of the lamin and LINC (LInker of Nucleoskeleton and Cytoskeleton) complex were altered in all these mutants. Whereas no significant differences were observed for ERK and AKT activities, the accumulation of DNA damage was associated to the *Lmna* p.R249W mutant myoblasts. Finally, significant myogenic differentiation defects were detected in the *Lmna* exon 4 mutants. These results have key implications in the development of future therapeutic strategies for the treatment of laminopathies.

## 1. Introduction

Lamins are nuclear intermediate filaments proteins that form a meshwork underlying the inner side of the nuclear membrane [1]. Nuclear lamins have been associated with a wide spectrum of cellular functions including structural support, mechanosensing, chromatin organization, transcription regulation, replication, nuclear assembly, and nuclear pore complex activity [2,3,4,5]. Laminopathies are a group of human rare diseases mainly associated with different mutations on the Lamin A/C gene (*LMNA*). Lamin A/C proteins are main components of the lamin. Laminopathies include at least 15 different diseases that are divided in 4 different categories depending on the affected tissues: striated muscle, adipose tissue, peripheral nerves, or multiple tissues [6]. This last group includes Hutchinson–Gilford progeria syndrome (HGPS), which is characterized by aging-like symptoms emerging in childhood.

The *LMNA* gene contains 12 exons and codes for 2 different proteins, Lamin A and C. Both are members of the intermediate filament group of proteins, which are characterized for the presence of a short N-terminal domain, a central domain composed of four α-helical domains that are separated by three linker regions and a globular C-terminal region. To date, more than 400 mutations in *LMNA* have been reported in patients (http://www.umd.be/LMNA/). These mutations are widely distributed throughout the entire gene. Moreover, there is no association between the gene location of these mutations and the wide range of phenotypes included in this group of rare diseases. Two main hypotheses have been postulated to explain the development and progression of laminopathies [7]. One of them assumes that *LMNA* mutations provoke structural and mechanical changes that are responsible for the development of the disease, whereas the other relies on functional abnormalities induced by changes in gene expression patterns and differentiation programs. Unfortunately, little is known about the exact mechanisms underlying the effects induced by the majority of the laminopathy-causing *LMNA* mutations.

No curative treatment is currently available for any type of laminopathy. A number of pre-clinical studies have explored the therapeutic potential of different compounds that modulate the Mechanistic Target Of Rapamycin Kinase (mTOR) pathway [8,9], the mitogen-activated protein kinase (MAPK) cascade [10,11], and the epigenetic regulator N-Acetyltransferase 10 (NAT10) [12], among others. Gene therapy approaches are being studied as well. In this regard, the most promising results have been obtained for HGPS by introducing frameshift mutations in the *LMNA* gene using CRISPR/Cas9 (Clustered Regularly Interspaced Short Palindromic Repeats/CRISPR associated protein 9) [13,14]. Besides these advances, we are still far from a cure for the majority of the laminopathies. More information is therefore needed about the molecular and cellular defects induced for each disease-associated *LMNA* mutation in order to obtain the therapeutic strategies that better fit each case.

Here, we explored whether mutations in *Lmna* exon 4 affect the biology and function of mouse myoblasts. This exon codes for a linker domain located between the two central α-helical domains of Lamins A and C (Appendix A). Although up to 47 mutations in *LMNA* exon 4 have been reported with different disorders linked to this gene (http://www.umd.be/LMNA/), little is known about the cellular and molecular changes induced by them that are causally related with laminopathy development and progression. Thus, although *LMNA* p.R249W is the most frequent mutation for LMNA-associated congenital muscular dystrophy, a severe type of laminopathy [15], and has been found to be related to abnormal nuclear morphology and myogenic differentiation, mislocalization of Lamin B, altered sense of microenvironment stiffness, and DNA damage [16,17,18,19], the causal connections between this mutation and the disease remain mainly unknown. On the other hand, because *LMNA* exons 3 and 5 are in frame, exon 4 skipping might be considered as a therapeutic strategy for all those laminopathies associated with *LMNA* exon 4 mutations. Although it has been reported that exon skipping could be successfully used for the treatment of diseases associated with *LMNA* mutations in exon 5 [20], the potential of this approach has not been tested yet for *LMNA* exon 4.

Using mouse C2C12 myoblasts and CRISPR/Cas9, we generated a collection of *Lmna* exon 4 mutants that were classified as *Lmna*-null, *Lmna*-R249W, or carriers of *Lmna* exon4 in-frame deletions. The study of these mutants allowed us to find critical information on the importance of *Lmna* exon 4 in myoblast biology, specifically in the integrity of the nucleus, the protein levels, and sub-cellular location of nuclear envelope proteins and the capacity to differentiate to myogenic fibers. Given the fact that *LMNA* mutations are causally associated with several rare diseases, this work provides new and valuable information for a better understanding and future treatment of these diseases with no cure.

## 2. Materials and Methods

Details of resources used in the research, including antibodies, cell culture media, reagents, and software, are listed in Table 1.

### 2.1. Cell Lines and 3D Model

For the majority of our experiments, we used C2C12 myoblasts, which are myoblasts originally established from satellite cells derived from the thigh muscle of a female C3H murine donor following a crush injury [21]. C2C12 cells and selected clones with mutations in exon 4 of *Lmna* have been grown at 37 °C of temperature and 5% CO_2_ conditions. The culture medium of myoblasts was based on Dulbecco’s modified Eagle’s medium (DMEM) supplemented with 10% fetal bovine serum (FBS) and 1% penicillin/streptomycin. Three human myoblast cell lines, previously described [18], were used in this work. Two were controls isolated from 25 and 38 year old healthy individuals (C25CL48 and AB1079, respectively). A third myoblast cell line was isolated from a 3 year old L-CMD patient who carried a *LMNA* p.R249W mutation. Human myoblasts were grown at a 37 °C and 5% CO_2_ conditions, in 1 volume of medium 199 with four volumes of DMEM with 20% FBS and 1% penicillin/streptomycin, and supplemented with 25 µg/mL fetuin, 5 ng/mL human epidermal growth factor (hEGF), 0.5 ng/mL basic fibroblast growth factor (bFGF), 5 µg/mL insulin, and 0.2 µg/mL dexamethasone.

3D muscle models were fabricated by photo-molding technique, as we have described previously [22] (Appendix A). Gelatin-based polymer solution (see Appendix B) was mixed with a C2C12 cell suspension to reach the final cell density of 25 × 10^6^ cell/mL. Briefly, a 6 µL drop of cell-laden polymer was placed on a functionalized coverslip (Appendix B), and a microstructured silicone (PDMS) stamp was pressed lightly on top, filling the microchannels with the solution. The sizes of PDMS stamp consisted of grooves of 100 µm, which were 100 µm in height with and ridges of 100 µm. The hydrogel was photo-crosslinked using a Ultraviolet Product (UVP) Crosslinker (Model CL-1000L, 365 nm, 40 W, from Analytik Jena U.S., Upland, CA, USA) by the exposure of 30 s under UV light. After carefully removing the stamp, the micro-structured cell-laden hydrogels were incubated for 1 day with culture medium. Due to high cell density in the hydrogel, growth medium was changed to differentiation medium, based in DMEM high glucose, supplemented with 2% horse serum and 1% penicillin/streptomycin, after 1 day to induce differentiation into myotubes.

### 2.2. Generation of Clones with Mutations in Lmna Exon 4 Using CRISPR/Cas9

All single guide RNAs (sgRNAs) were designed to target exon 4 of the *LMNA* gene using the Breaking Cas Design tool (https://bioinfogp.cnb.csic.es/tools/breakingcas/) [23]. Three sgRNAs for the *Lmna* exon 4 (sg10, sg11, and sg12; Table 2) were cloned in the pX459 vector. The C2C12 cells were transfected, using Lipofectamine 3000, with the pX459-sgRNA vector with the corresponding template (ssODN) in low (30 pmol) or high (300 pmol) doses. Templates included forms mutated only in the Protospacer Adjacent Motif (PAM) (mut) or mutated in the PAM and in the Protospacer (mut2) (Table 2). After 48 hours of transfection, puromycin (2 µg/mL) was added for 5 days. The selected cells were grown until a pool was obtained per condition. The clones were picked up after seeding at low density of the pools. The pools and clones were expanded to allow the complete analysis. For the control samples, we followed the same process but using an empty px459 vector and no templates.

### 2.3. DNA Sequencing and Bioinformatic Analysis

To identify mutations at *Lmna* exon 4, MiSeq DNA sequencing was carried out in the Genomic Unit of Instituto de Salud Carlos III. DNA was isolated from the different pools and clones. Next, Illumina adapters were added to the target region of the Cas9 nuclease by PCR using the DeepSeq-Fw and DeepSeq-Rv primers (Table 2).

The resulting product was used for a second PCR, where the specific indexes were added to every sample. Genomic reads in FASTQ format were analyzed using the CRISPResso platform. This software isolates the part of each read spanning the chosen region, highlights small insertions/deletions, and outputs a count of each regional sequence. We then analyzed the percentage of the sequences showing regional differences in control and with the different sgRNA-exon 4 of *LMNA*-transduced samples. All the pools and clones were also analyzed by Sanger sequencing of PCR products amplified using *Lmna* exon 4-specific primers Sanger-mLmna_Ex4_Fw and Sanger-mLmna_Ex4_Rv (Table 2). Sanger sequences were analyzed using the TIDE platform.

### 2.4. Western Blot Analysis

SDS-PAGE was performed using standard procedures. Briefly, cells were disrupted with RadioImmunoPrecipitation Assay (RIPA) buffer (150 mM NaCl, 1% Nonyl Phenoxypolyethoxylethanol (NP-40), 0.5% sodium deoxycholate, 0.1% SDS, 50 mM Tris-HCl (pH 7.5), and protease and phosphatase inhibitors). The lysates were sonicated with the ultrasonic processor UPH100H (Hielscher, Teltow, Germany) (three pulses of 5 s at 100% amplitude) to allow dissociation of proteins. Samples were centrifuged at 14,000 rpm at 4 °C for 15 min. Protein concentration was quantified with the BCA system. Table 1 lists the references and dilutions of the primary and Horseradish Peroxidase (HRP)-labelled secondary antibodies used for Western blot analyses. Detection was carried out using an ECL western blotting system.

### 2.5. Immunofluorescence Microscopy, Nuclear Morphology Analyses, and Myogenic Differentiation Studies

For 2D cultures, cells plated onto coverslips were fixed with methanol for 5 min at −20 °C. Cells were incubated in blocking solution (0.1% bovine serum albumin (BSA), 1% FBS, and 1% horse serum in phosphate-buffered saline (PBS), in the case of SUN2; 1% BSA, 2.5% goat serum, 2.5% donkey serum, and 0.3% Triton X-100 in PBS, in the cases of lamin A/C, lamin B1, and emerin) for 30 min. Primary and secondary antibodies used for immunofluorescence detection are detailed in Table 1. Coverslips were mounted using Prolong Gold (ThermoFisher Scientific; Waltham, MA, USA) with DAPI, and images were captured using a Leica TCS SP5 confocal microscope (Leica, Wetzlar, Germany) and a 63× (HCX PL APO 63× 1.4 OIL).

For the 3D models, cells were fixed in a 10% formalin solution (Sigma-Aldrich, St. Louis, MI, USA). Then, hydrogels were washed with Tris-buffered saline (TBS), permeabilized with 0.1% Triton X-100 in TBS for 15 min, and blocked with a blocking buffer consisting of 0.3% Triton X-100 and 3% donkey serum in TBS for 2 h. Afterwards, samples were washed with TBS and incubated with anti-MYH7 in blocking buffer (O/N, 4 °C). Then, samples were washed with TBS and incubated with rhodamine phalloidin and goat anti-rabbit A488 in blocking buffer (O/N, 4 °C). After washing with TBS, nuclei were counterstained with 1 µM DAPI for 15 min.

Cells (5000 cells per well) were plated into a 96-well cell culture black microplate with a glass bottom (Greiner Bio-one, Kremsmünster, Austria). After 24 h, the cells were fixed with methanol for 5 min at −20 °C. The fixed cells were incubated with 2 µL/mL Hoechst 33324 in PBS for 10 min at 37 °C. Finally, nuclear morphology was evaluated by determining the circularity index using the Cytell Cell Imaging System (GE Healthcare Life Sciences, Marlborough, MA, USA). The circularity values were between 0 and 1, with 1 being the value of a perfect circle. For the nuclei morphology in the 3D muscle model, confocal microscopy images were analyzed using ImageJ software (National Institutes of Health and the Laboratory for Optical and Computational Instrumentation, Madison, WI, USA).

Myoblast differentiation was induced by growth medium exchange for the differentiation medium (DMEM with 2% horse serum and 1% of penicillin/streptomycin). The cells were differentiated for 5 days. The cells (1 × 10^5^ cells per well) were fixed with methanol for 5 min at −20°C, permeabilized with 0.05% Triton X-100 in PBS for 5 min at room temperature, and washed with PBS for 5 min. Then, coverslips were blocked in blocking solution (15% FBS in PBS) for 30 min at room temperature. The cells were incubated with primary antibody against myosin heavy chain (MF20) for 1 h at room temperature. The cells were washed with PBS three times for 5 min, incubated with secondary antibody anti-mouse Ig for 45 min at room temperature and in the dark, and washed with PBS three times for 5 min. Coverslips were mounted using Prolong Gold (Thermo Fisher Scientific) with DAPI. Stained myotubes were visualized in a Leica DM4B microscope (Leica-Microsystems, Wetzlar, Germany) using the 10× (N PLAN 10×/0.25 DRY) and 20× objectives (HC PL FLUOTAR 20×/0.55 DRY).

### 2.6. Quantification and Statistical Analysis

The levels of Lamin A/C, Lamin B1, Emerin, SUN1, and SUN2 at the nucleus were analyzed using ImageJ software (National Institutes of Health). In brief, individual lines of constant length were plotted perpendicular to the nucleus and used to evaluate peak fluorescence intensity using the PlotProfile feature from ImageJ. Random individual plot profiles of conditions were generated and quantified at the nuclei using the PlotProfile feature from ImageJ. In every set of experiments, all immunofluorescence images were taken from single confocal acquisitions under the same exposure conditions.

All the graphs and statistical analyses were carried out with GraphPad 8.0 software (GraphPad, San Diego, CA, USA). Error bars represent standard error of the mean (SEM). For statistical analyses, the level of significance was determined by two-tailed, unpaired Mann–Whitney *U* test or Student’s *t*-test.

## 3. Results

### 3.1. Generation of Lmna Exon 4 Mutations in C2C12 Cells

In order to produce *Lmna* exon 4 mutants by CRISPR/Cas9 technology, we designed three sgRNAs to target the Cas9 endonuclease to this *Lmna* region (Figure 1A). Following the scheme depicted in Figure 1B and detailed in Section 2, we produced a total of 18 pools and 210 clones in which CRISPR/Cas9 activity (insertions and deletions, Indels) was measured by TIDE and deep sequence analyses (Appendix A). It is interesting to note that each sgRNA induced a different Indel pattern (Appendix A). One single nucleotide deletion was the most frequent event when using sg12 (present in 77% of the clones), whereas a wider spectrum of deletions and insertions was observed for sg10 and sg11 guides. Interestingly, deletions of 3 and 6 nucleotides, which produce in-frame deletions mutants, were mainly observed in sg11-transfected cells (they were detected in 20% and 35% of the clones, respectively). Sequences with no Indels were detected in 3%, 5%, and 23% of the clones transfected with s10, sg11, and sg12 guides, respectively.

We also aimed at introducing the R249W mutation, one of the human variants associated with a severe type of laminopathy, *LMNA*-associated congenital muscular dystrophy. In all the transfections, we incorporated a donor template including the mutant nucleotide. Although five clones were positive for the R249W mutation, only one (g11H-D6; Table 3) was free of additional nucleotide changes and, therefore, allowed the expression of this mutant form.

A total of 113 clones, named as “null”, only carried frameshift mutations that produced truncated proteins due to premature stop codons. Sixteen clones, classified as “delta”, were characterized by the only presence of triplet deletions compatible with the formation of Lamin A/C proteins with deletions of 1, 7, 14, 15, or 57 amino acids (Appendix A). Ten clones that were transfected with Cas9 expressing vectors with no sgRNAs showed no mutations in *Lmna* exon 4 and were used and named as “controls”. Table 3 lists the control, null, delta, and R249W clones that were selected for a detailed study on the effects caused by the mutations in *Lmna* exon 4.

### 3.2. Components of the Nuclear Lamina Were Abnormally Distributed in Lmna Exon 4 Mutant Myoblasts, Lamin A/C Abnormal Expression, and Subcellular Localization in Lmna Exon 4 Mutants

We first analyzed the total protein levels and nuclear sub-cellular localization of Lamin A/C. As shown in Figure 2A, all the *Lmna* exon 4 mutants were characterized by a significant reduction in the amount of both Lamin A and C proteins. This feature was accompanied by an abnormal subcellular localization of Lamin A/C in all delta and R249W mutants (Figure 2B). Whereas control and R249W myoblasts showed a predominant Lamin A/C signal in the nuclear periphery, delta clones showed no differences in Lamin A/C intensity between perinuclear and nucleoplasmic regions. Moreover, in the different delta clones, we were able to detect a wide spectrum of abnormalities regarding Lamin A/C staining including honeycomb-like patterns, asymmetric distribution, capping, and bright foci (Appendix A). Although Lamin A/C subcellular localization in R249W cells followed the same pattern found in control myoblasts, a detailed analysis showed lower fluorescence intensity of Lamin A/C at the nuclear periphery when compared with the control clones (Figure 2B).

To investigate the consequences of the Lamin A/C mislocalization and low expression in *Lmna* exon 4 mutants, we studied, using single confocal imaging, the sub-cellular localization of Lamin B1 and Emerin, two main inner nuclear membrane proteins. In control clones, Lamin B1 accumulated at the nuclear periphery and was also detected, to a lesser extent, at the nucleoplasm. Lamin B1 distribution was slightly altered in *Lmna* exon 4 mutants, which did not show a sharp contrast between the Lamin B1 signal detected in Lamin area and nucleoplasm (Figure 2B). A significant effect was also expected for Emerin because the localization of this protein is dependent on Lamin A/C [24]. Indeed, Emerin sub-cellular distribution was affected in null, delta, and R249W clones (Figure 2B). In all the cases, Emerin extranuclear signal was compatible with the endoplasmic reticulum pattern previously reported for *Lmna* null and mutant cells [25,26,27].

Our results indicated that a reduction and abnormal sub-cellular localization of the main lamina components—Lamin A/C, Lamin B1, and Emerin—was common to all the *Lmna* exon 4 mutants analyzed.

### 3.3. Nuclear Morphology Abnormalities Were Common for All the Clones with Mutations in Lmna Exon 4

The abnormalities described for all the *Lmna* exon 4 mutants were compatible with a defective lamin that might affect the nuclear membrane integrity. To test this hypothesis, we investigated the nuclear morphology of all the clones analyzed in this work. Whereas nuclei of controls are circular or ovoid, mutant clones are characterized by irregular nuclei with multiple herniations and, more frequently, by elongated nuclei. To quantify these differences in terms of nuclear morphology, we calculated the circularity index of the nuclei of myoblasts asynchronously growing in 2D cultures. As expected, all the mutants showed a significant reduction in the circularity index of their nuclei (Figure 3A).

To study nuclear morphology in a more physiologic condition, we encapsulated the myoblasts in a mimetic 3D architecture that recreated a native skeletal muscle (Appendix A). Under this 3D cellular organization, all type of clones formed highly elongated and aligned fibers (Appendix A). For both control and mutant clones, the circularity was reduced when compared to the values obtained in 2D experiments (Figure 3B). This effect was probably caused by the geometric constriction of the pattern and the compliant nature of the hydrogel used to create the 3D structure. Importantly, circularity was significantly reduced in mutant clones when they were compared to controls (Figure 3B).

These results demonstrated that mutations in *Lmna* exon 4 negatively affect nuclear morphology in myoblasts growing in both 3D and 2D conditions.

### 3.4. Abnormal Myogenic Differentiation Was Common to All Myoblasts Carrying Lmna Exon 4 Mutations

*LMNA* mutations have been associated not only with structural abnormalities, such as the nuclear integrity defects described above, but also with functional deficiencies, including differentiation capacity to myogenic fibers [28,29]. To test if this was also the case for the mutants generated in this work, we induced myogenic differentiation in all the myoblasts under study. In a first set of experiments, we studied the differentiation capacity of control and mutant clones growing in 2D conditions. As shown in Figure 4A, 5 days post-induction of differentiation, control myoblasts formed myogenic fibers that were positive for myosin heavy chain (MHC). However, in null, R249W and the majority of the delta clones, we were not able to detect either myogenic fibers or MHC expression. Interestingly, delta57 clone, which is missing the whole *Lmna* exon 4, showed a moderate formation of fibers and MHC expression.

Similar results were obtained from myoblasts growing in 3D structures (Figure 4B). In this case, MHC was only detected in control clones as well as in delta57 and R249W mutants. Importantly, although R249W cells were positive for MHC, they showed an abnormal distribution of this protein that could indicate the formation of abnormal myogenic fibers.

Our results indicate that *Lmna* exon 4 mutations significantly affect the capacity of C2C12 cells to differentiate to myogenic fibers.

### 3.5. Expression of Lmna Exon 4 Mutants Was Associated with Abnormal Subcellular Localization of SUN1 and SUN2

In an effort to understand the mechanisms underlying the nuclear morphology and myogenic differentiation defects that characterize the *Lmna* exon 4 mutants, we studied the status of SUN1 and SUN2. These proteins, together with Lamin A/C, Emerin, and Nesprins, among others, constitute the LINC complex, a molecular link that connects the nucleus to the cyto-skeleton, and have been associated with muscular dystrophies [30]. The total protein levels of SUN1 and SUN2 were not altered in any of the mutant types in null, delta, or R249W (Figure 5A). However, their subcellular distribution was significantly affected in all the mutants when compared with controls (Figure 5B). Whereas SUN1 and SUN2 accumulated in the nuclear envelope of control myoblasts, they were barely detected in the same region of *Lmna* exon 4 mutants.

These results, together with the ones obtained for the localization of Emerin and Lamin proteins, indicated that the defects induced at the nuclear envelope by the *Lmna* exon 4 mutations significantly impact the subcellular localization of LINC complex components.

### 3.6. MAPK Signaling Was Not Altered in Lmna Exon 4 Mutant Myoblasts

Thanks to the analysis of the *Lmna^H222P^* mouse model, a connection between *LMNA* mutations and signaling pathways has been discovered [31]. Specifically, enhanced phosphorylation of ERK1/2 and AKT has been reported in the heart of *Lmna^H222P^* mice [10,32]. Therefore, we decided to study the status of these key signaling molecules in *Lmna* exon 4 mutants. Although the activation of ERK1/2, estimated by the quantification of phosphorylated-ERK1/2, was very heterogeneous for inter and intra clones (Appendix A), no significant differences were observed when control, null, delta, and R249W groups were compared (Figure 6A). Likewise, there were no significant differences in AKT activation (Figure 6B).

Therefore, the *Lmna* exon 4 mutations studied here were not characterized by homogenous and significant changes in the activation of the signaling proteins ERK1/2 and AKT.

### 3.7. High Levels of DNA Damage Were Associated with R249W Mutation

We also wanted to study the effect of *Lmna* exon 4 mutations at the DNA level, as progeroid and striated muscle laminopathies have been associated with high levels of DNA damage [19,33,34,35]. Quantification of DNA damage by detection of γH2AX (phosphor-H2A.X Variant Histone) in asynchronously growing myoblasts revealed elevated levels of DNA damage in the R249W clone, whereas no signal was detected in the rest of mutants (null and delta) or controls (Figure 7A). To further analyze the connection between the R249W mutation and DNA damage, we also explored it in human myoblasts carrying a *LMNA* p.R249W mutation causing L-CMD. Interestingly, the human R249W mutant showed a clear increase in γH2AX levels when compared with the controls (Figure 7B).

These results indicate that high levels of DNA damage are a specific feature of *LMNA* p.R249W myoblasts.

## 4. Discussion

### 4.1. Comparison of Lmna Exon 4 Mutants Generated in This Study with Previously Reported LMNA Mutants

In this study, using CRISPR/Cas9 technology, we analyzed a collection of *Lmna* exon 4 mutants in C2C12 mouse myoblasts. Three different types of mutants were generated: null mutants, due to frameshift mutations; delta mutants, caused by in-frame deletions; and an already known R249W mutant. Interestingly, none of the delta and null mutants generated in this work have been previously reported in patients. Upon studying a number of molecular, cellular, and functional features, we can conclude that the subcellular localization of key members of the LINC complex and nuclear envelope, including Lamin A/C, Lamin B1, Emerin, SUN1, and SUN2, as well as the nuclear membrane integrity and myogenic differentiation capacity, were negatively affected.

The comparison of the features of the *Lmna* mutants described here with those previously reported by others resulted in coincidences and differences. For instance, the Emerin extranuclear localization, compatible with endoplasmic reticulum, detected in null, delta, and R249W clones, is similar to the previously reported for *Lmna* null, *Lmna^L530P/L530P^*, *Lmna^dK32^*, and *Lmna^N195K^* mouse myoblasts, as well as *LMNA* p.Y259X patient cells [15,25,26,27,36]. Contradictory results have been described for Lamin B1 localization. It has been reported that the loss of A-type lamins allows the assembly of B-type lamins [25]. In addition, more recently, LMNB1 localization defects have been shown in induced pluripotent stem cells carrying *LMNA* mutations [17], which is in line with the results we obtained in the present work. On the other hand, the *Lmna* null, delta, and R249W analyzed in this work showed a nuclear, although low intensity, signal for SUN1 and SUN2 at the nuclear envelope. A regulation by Lamin A of SUN2 localization has been previously reported in mouse fibroblasts [37]. In addition, SUN2 nuclear distribution has been dramatically altered in synaptic nuclei of *Lmna^−/−^* and *Lmna^H222P/H222P^* muscle [38]. On the other hand, the weak nuclear envelope localization characteristic for *Lmna* exon 4 mutant C2C12 cells differs from initial reports showing that localization of SUN1 does not depend on A-type lamins in mouse fibroblasts and HeLa cells [38,39,40]. However, other works in *Lmna* null or *Lmna^L530P/L530P^* mouse fibroblasts showed an extranuclear signal or loss of nuclear envelope signal [27,41,42]. The differences observed between these reports and the present work might be due to the different cell system used—fibroblasts versus myoblasts. Importantly, it has been reported that *LMNA* mutations causing Emery–Dreifuss muscular dystrophy are associated with low SUN1 levels in human myoblasts [43], an outcome similar to the one obtained in the *Lmna* exon 4 mutants generated and characterized in this work. In summary, different conclusions might be obtained for the SUN1–Lamin A/C connection that could be a consequence of different cellular or molecular scenarios. Further studies are needed to better clarify the role that Lamin A/C proteins play on SUN1 localization and function.

*LMNA* p.H222P, one of the most studied *LMNA* mutations, is also located in exon 4. However, some of its features differ from the data obtained with the exon 4 mutants analyzed here. For instance, the alterations of MAPK and AKT signaling pathways described in the *Lmna^H222P–H222P^* model [10,32] were not detected in the *Lmna* exon 4 mutants analyzed in this work. Moreover, *Lmna* knockdown in C2C12 cells has been previously associated with a significant increase in ERK1/2 activation [44], an abnormality that is not found in the null C2C12 clones studied here (Figure 6A). To note, the siRNA (small interfering Ribonucleic Acid) used in the former did not fully inhibit Lamin A/C expression, whereas the use CRISPR/Cas9 induced the complete elimination of Lamin A/C expression (Figure 2A). In our study, we observed a high heterogeneity both inter and intra clones for p-ERK and p-AKT levels. Because this was not the case for other proteins analyzed here (Appendix A), we conclude that none of these two pathways are homogeneously altered in the *Lmna* exon 4 mutants studied in this work, whereas *Lmna* p.H222P mutation is specifically associated with an increased activation of both pathways. Another possible explanation, would be the fact that the majority of the studies for *Lmna* p.H222P have been carried out using cardiomyocytes from a *Lmna^H222P^* knockin mouse model, whereas the cells used in the present study were mouse myoblasts. The detailed characterization of the same cell type from a mouse *Lmna^R249W^* model (currently under study) will help us to discern the correct explanation of these differences.

### 4.2. Mechanistic and Functional Defects Associated with Lmna Exon 4 Mutations

Myogenic differentiation is the most important function of myoblasts. It has been previously shown that the expression of Lamin A mutants associated with different laminopathies inhibits the in vitro differentiation of C2C12 myoblasts [28,29]. Abnormal myoblast differentiation has also been reported in *Lmna^dK32^* cells [15]. Moreover, *Lmna* null satellite cells are characterized by a defective myogenic differentiation [45]. The same defects were observed for the majority of the null, delta, and R249W mutants analyzed in this work, which confirmed the fact that mutations and/or downregulation of Lamin A/C are associated with an impairment in myogenic differentiation that might explain a significant part of the pathological phenotype of laminopathies. However, other scenarios cannot be ruled out. For instance, a premature differentiation associated to SMAD6 (SMAD Family Member 6) overexpression and BMP4 (Bone Morphogenetic Protein 4) downregulation has been reported in *Lmna^Δ8–11^* myoblasts [46]. Although this seems not to be the case for *Lmna* exon 4 mutant myoblasts (data not shown), further studies will help to fully understand the differentiation problems induced by these *Lmna* mutations.

Other potential mechanisms underlying the abnormal differentiation of the *Lmna* exon 4 mutants is an abnormal function of the LINC complex. The deficient subcellular localization observed for many of the LINC complex components (Emerin, Lamin A/C, Lamin B1, and SUN1/SUN2) in *Lmna* exon 4 mutant myoblasts could be the cause of the defects detected in nuclear membrane integrity and myogenic differentiation. Interestingly, the LINC complex, a molecular link between the nucleo- and cyto-skeleton, has been associated with muscular dystrophies [30]. Furthermore, the alteration of the nuclear transport due to a malfunction of the nuclear pores has already been associated with LINC complexes defects [47,48], and might be the functional cause of the pathological phenotype associated with *Lmna* exon 4 mutants. However, additional studies are needed to confirm the causal relationship between *Lmna* exon 4 mutations and LINC complex in laminopathies.

Other mechanisms and pathways previously associated with muscular dystrophies and cardiomyopathies might be associated with the *Lmna* exon 4 mutations. For instance, an altered WNT (Wingless and Int-1)/β-catenin signaling has been found in hearts of *Lmna^H222P^* mice [10,11], and pharmacological activation of this pathway improves heart dysfunction in this mouse model [49]. On the other hand, disruption of the nucleocytoplasmic Ran gradient and consequent nuclear protein import defects have been reported in cells carrying LMNA mutations [5,50]. Moreover, alterations of the nuclear morphology, such as the ones associated with *Lmna* exon 4 mutants, are associated with an abnormal distribution of the chromatin and, subsequently, changes in the epigenetics marks [51,52]. Whether all these pathways are also causally connected with the *Lmna* exon 4 mutations we have described herein remains to be determined and will be part of further studies.

### 4.3. DNA Damage: A Feature Specific for the R249W Mutation?

The nuclear lamina is a critical structure for the stability and proper nuclear distribution of DNA [53]. A weak nuclear envelope affects DNA integrity, inducing the accumulation of DNA damage, a feature linked to *LMNA*-associated diseases [19,54,55,56]. The majority of the mouse myoblasts studied in this work did not show elevated levels of DNA damage, which is consistent with the fact that we have explored this property only in undifferentiated myoblasts, and increased levels DNA damage are mainly observed in differentiated cells and tissues of HGPS and *LMNA*-related, muscular dystrophies [19,55]. Surprisingly, even in this undifferentiated myoblast state, the *Lmna* p.R249W mutant under study showed significantly elevated levels of DNA damage. The fact that this feature was also observed in human *LMNA* p.R249W myoblasts (Figure 7) and in muscle biopsies from a 1 year old patient carrying a *LMNA* p.R249W mutation [19I] points to a specific association with DNA integrity and/or repair mechanisms. Further studies will help to validate this connection and explore therapeutic opportunities related to DNA damage and repair.

### 4.4. CRISPR/Cas9 Activity and Gene Therapy Implications

The delta and R249W clones analyzed in this work showed a significant reduction in the levels of Lamin A/C (Figure 2). This was probably due to the CRISPR-mediated deletion of one or more *Lmna* alleles (C2C12 cells used in this work were hypertriploid), as mRNA expression was significantly reduced in all of them (data not shown). Therefore, the phenotypes observed in the *Lmna* exon 4 mutants could have been due not only to the mutations generated in *Lmna*, but also to a defective expression of lamin proteins. It is important to note that *Lmna* haploinsufficiency is associated with dilated cardiomyopathy [57]. On the other hand, all the clones analyzed in this work lost their wild type alleles and showed a single mutant allele. This genomic scenario strongly differs from the predominantly autosomal dominant scenario of LMNA-associated diseases in which missense mutations could lead to Lamin or Prelamin A accumulation. To note, a completely different situation is found in the null mutations generated by CRISPR/Cas9. All this evidence has important implications for the therapeutic potential that CRISPR technology could have for the treatment of laminopathies and the potential use as disease models of the clones we generated.

One of the most interesting results we obtained in this work was the high percentage of delta clones isolated upon CRISPR/Cas9 activity on *Lmna* exon 4 (Appendix A). This indicated that these types of in-frame mutations might have some selective advantage over the rest of the Indels generated by CRISPR/Cas9 activity. Moreover, one of these clones, delta57, is characterized by the expression of a *Lmna* exon 4 null gene. Interestingly, although delta57 shows nuclear morphology and LINC complex abnormalities, it does retain some myogenic differentiation capacity. This could indicate that an exon 4 skipping strategy for the elimination of mutations located in this exon could be compatible with a partially functional myoblast. As far as we know, in *LMNA*-associated disease, successful exon skipping has only been shown for exon 5 in human cells [20]. The potential of this therapeutic strategy needs further experimentation in appropriate in vivo models.

In conclusion, in this work, we characterized in detail the consequences associated with mutations in *Lmna* exon 4 in myoblasts. This new knowledge has opened the possibility of exploring new therapeutic approaches based on the DNA damage specifically induced by the *Lmna* R249W mutation and the use of exon skipping strategies for the elimination of *Lmna* exon 4 mutations.

## Figures and Tables

**Figure 1 cells-09-01286-f001:**
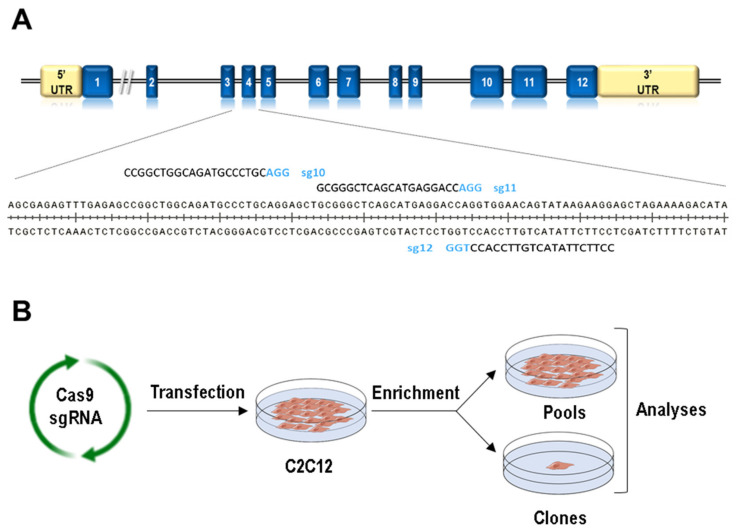
Generation of Lamin A/C gene (*Lmna*) exon 4 mutations using CRISPR/Cas technology. (**A**) Scheme for exon 4 *Lmna* sgRNA localization. (**B**) Work-flow followed for the generation of C2C12 cells carrying mutations in *Lmna* exon 4.

**Figure 2 cells-09-01286-f002:**
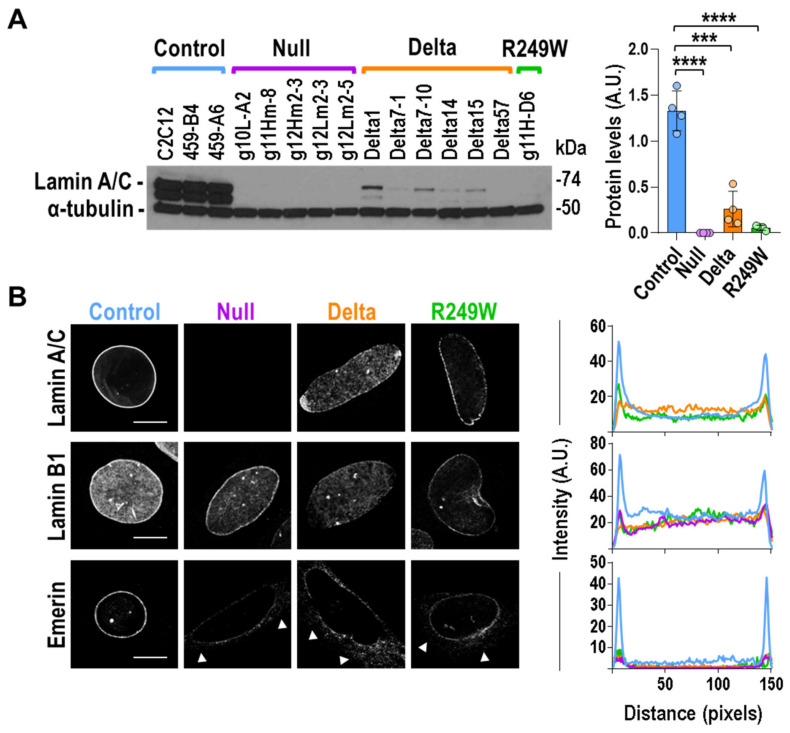
Components of the nuclear lamin were abnormally distributed in *Lmna* exon 4 mutant myoblasts. (**A**) Lamin A and C levels were reduced in *Lmna* exon 4 mutants. Quantification of Lamin A and C protein levels. Left pictures show one of the three independent biological replicates analyzed. The other replicates are included in Appendix A. Quantification of Lamin A/C signal in the three independent analyses, by mutant type, is shown on the right. Bars are mean ± SEM. Two-tailed, unpaired Student’s *t*-test: **** *p* < 0.0001, *** *p* = 0.003, *n* = 4. (**B**) Mouse myoblasts were immunostained for Lamin A/C, Lamin B1, and Emerin. Representative single confocal images from nuclei are shown for each *Lmna* group. Extra-nuclear signal was observed for Emerin in null, delta, and R249W mutants (arrows). Scale bar = 8 μm. Right plots show average fluorescence intensity across nuclei by mutant type (control (blue): three clones; null (purple): five clones; delta (orange): six clones; R249W (green): one clone; *n* = 5 nuclei per clone). Peaks at both ends of the plot represent peripheral staining. Representative images from other clones are shown in Appendix A (Lamin A/C), Appendix A (Lamin B1), and Appendix A (Emerin).

**Figure 3 cells-09-01286-f003:**
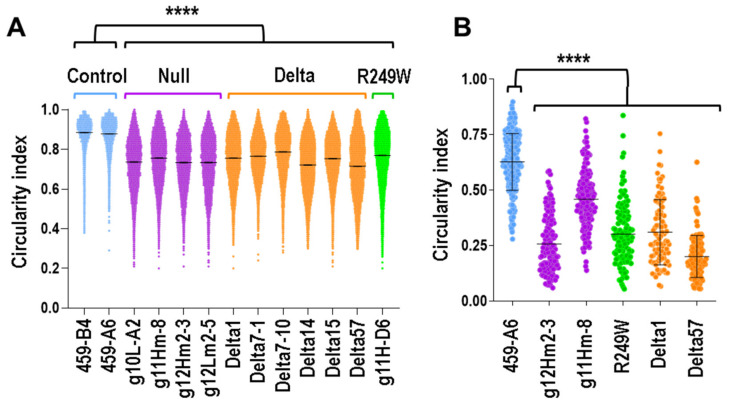
Nuclear morphology was altered in mouse myoblasts carrying *Lmna* exon 4 mutations. (**A**) Circularity index was calculated in nuclei from clones asynchronously growing in 2D cultures (**** *p* < 0.0001; *n* > 5000 nuclei per clone). (**B**) Plot shows the circularity index from clones encapsulated in a 3D pattern. Data were compared using an unpaired Mann–Whitney *U* test (**** *p* < 0.0001). Blue, purple, green, and orange colors indicate control, null, R249W, and delta clones, respectively.

**Figure 4 cells-09-01286-f004:**
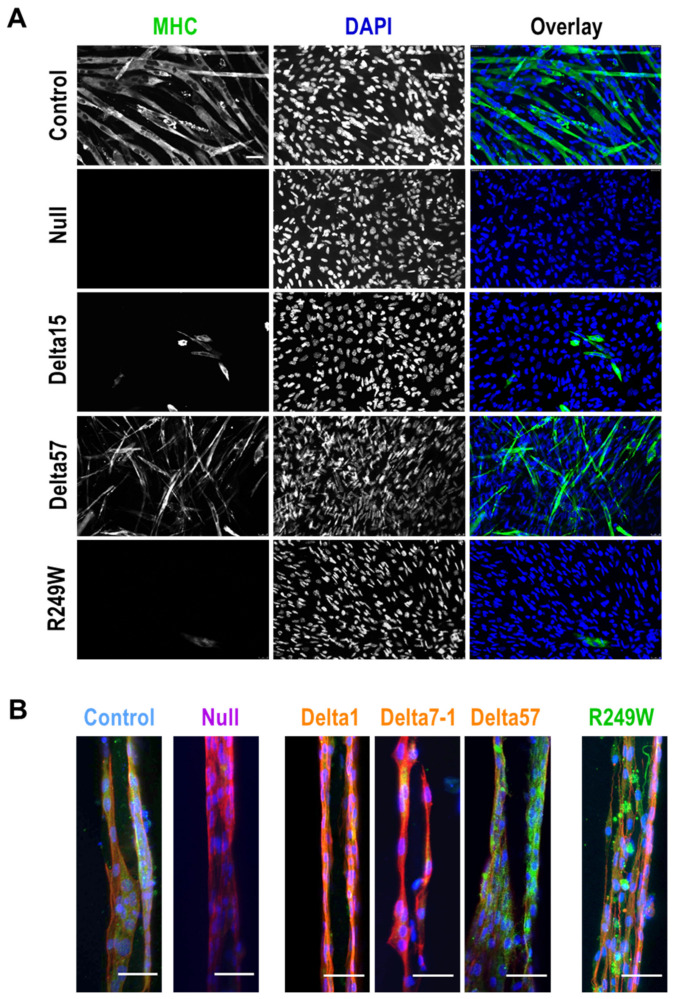
Myogenic differentiation was impaired in mouse myoblasts with mutations in *Lmna* exon 4. (**A**) Myoblasts growing in 2D conditions were induced to differentiate to myogenic fiber as described in Section 2. Representative pictures show myosin heavy chain (MHC, green) and nuclei (DAPI, blue). Scale bar = 60 µm. (**B**) Myogenic fiber formation was also affected in 3D models. Representative pictures show, at 7 days post-differentiation, myosin heavy chain 7 (green), nuclei (blue), and phalloidin staining (red). Control and null samples correspond to 459-A6 and g12Lm2-3 clones. Scale bar = 50 µm. Single color channels for this panel are shown in Appendix A.

**Figure 5 cells-09-01286-f005:**
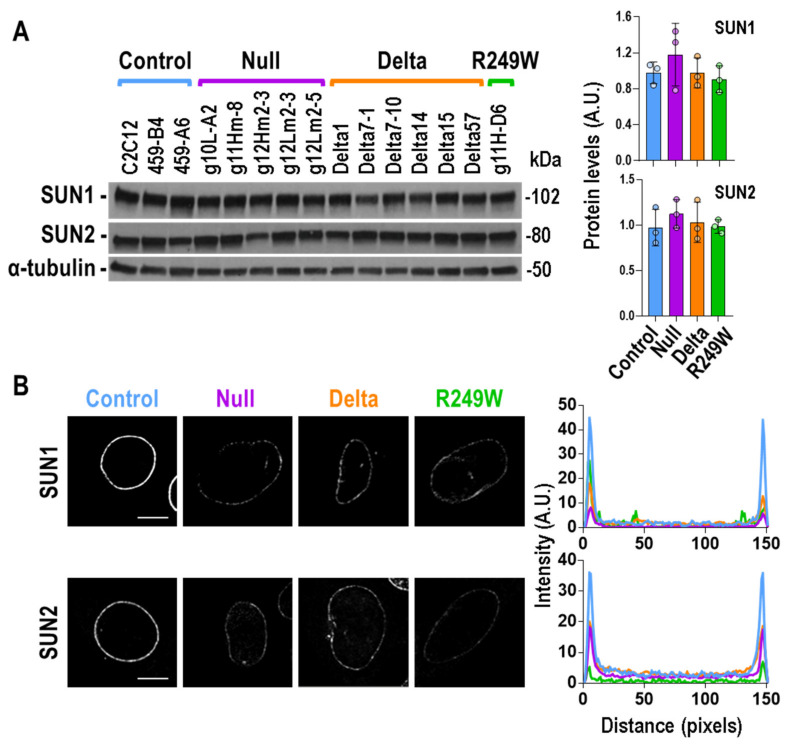
SUN1 and SUN2 protein levels and sub-cellular localization in *Lmna* exon 4 mutant myoblasts. (**A**) Total amount of SUN1 and SUN2 proteins was determined by Western blot. One of the three biological replicates is shown. The rest of the Western blots are shown in Appendix A. Quantification, by mutant type, of SUN1 and SUN2 signals for the three independent analyses is shown in the graphs on the right. Bars are mean ± SEM, *n* = 3. No significant differences were detected when mutants were compared with controls using a two-tailed, unpaired Student’s *t*-test. Representative, single, confocal, images of SUN1 (**B**) and SUN2 (**C**) detected by immunostaining are shown. A decreased nuclear envelope signal was observed in null, delta, and R249W mutants. Scale bar = 8 μm. Right plots show average fluorescence intensity across nuclei by mutant type (control (blue): three clones; null (purple): five clones; delta (orange): six clones; R249W (green): one clone; *n* = 5 nuclei per clone). Peaks at both ends of the plot represent peripheral staining. Representative images for SUN1 and SUN2 in all the clones are shown in Appendix A, respectively.

**Figure 6 cells-09-01286-f006:**
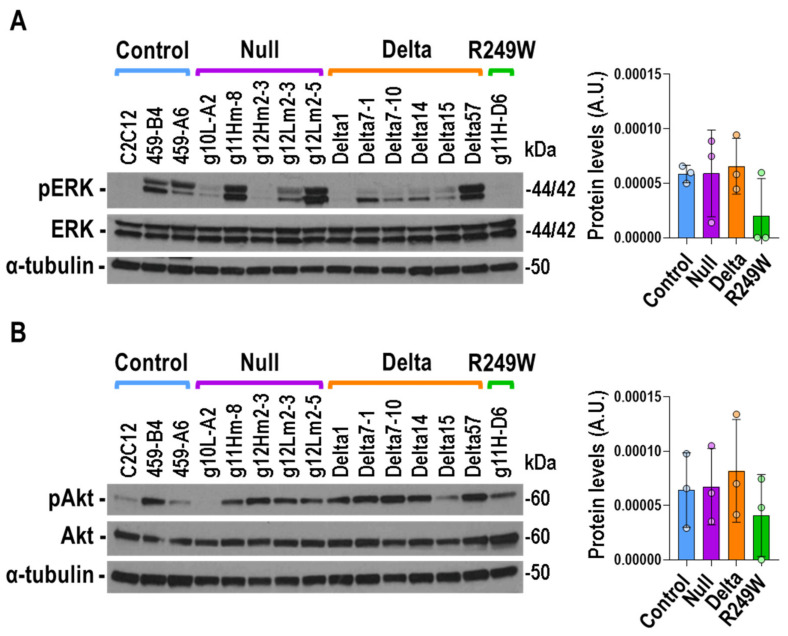
No differences in ERK1/2 and AKT activation were detected in *Lmna* exon 4 mutant myoblasts. (**A**) Representative immunoblot for the detection of phosphorylated (pERK) and total (ERK) ERK1/2 in *Lmna* exon 4 mutants. (**B**) Representative immunoblot for the detection of phosphorylated (pAKT) and total (AKT) AKT. Detection of α-tubulin was used in all the cases as loading control. Data in bar graphs are mean ± SEM of all the clones of each type, *n* = 3. No significant differences were detected when mutants were compared with controls using a two-tailed, unpaired Student’s *t*-test. The remainder of the Western blots analyzed in (**A**) and (**B**) are shown in Appendix A.

**Figure 7 cells-09-01286-f007:**
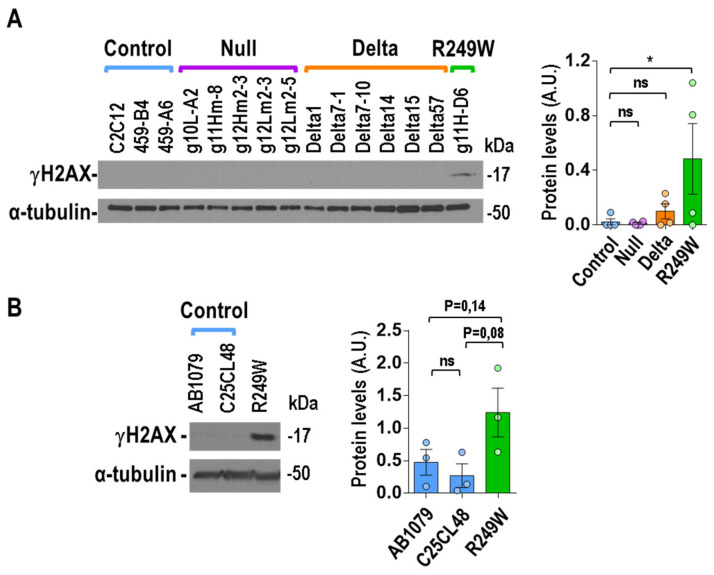
High levels of DNA damage were specifically associated with *LMNA* R249W mutation. (**A**) Representative immunoblot showing γH2AX levels in *Lmna* exon 4 mutant myoblasts. (**B**) Immunoblot showing γH2AX levels in human myoblasts from two controls and one patient carrying the p.R249W mutation. Detection of α-tubulin was used in all the cases as loading control. Data in bar graphs are mean ± SEM, *n* = 3. In (**A**) and (**B**), data were compared using a two-tailed, unpaired Student’s *t*-test. * *p* < 0.05. The remainder of the Western blots analyzed in (**A**) and (**B**) are shown in Appendix A.

**Table 1 cells-09-01286-t001:** Key resources used in this work.

ANTIBODIES			
Name	Source (identifier)	Western Blot	Immuno- Fluorescence
Anti-mouse Lamin A/C (E-1)	Santa Cruz Biotechnology (Dallas, Texas, USA) (sc-376248)	1:3000	1:500
Anti-mouse α-tubulin	Sigma-Aldrich (St. Louis, MI, USA) (T9026)	1:5000	
Anti-rabbit lamin B1	Abcam (Cambridge, UK) (ab16048)		1:100
Anti-rabbit emerin (D3B9G)	Cell Signaling (Danvers, MA, USA) (#30853)		1:100
Anti-mouse Sad1 And UNC84 Domain Containing 1 (SUN1) BBmSun1 IgC2b X12.11	Donated by Dr. Colin Stewart	1:50	1:50
Anti-rabbit SUN2 (Sad1 And UNC84 Domain Containing 2) 11905	Donated by Dr. Eric Schirmer	1:500	1:200
Anti-rabbit p44/42 Mitogen-Activated Protein Kinase 1/3 (Erk1/2)-137F5	Cell Signaling (Danvers, MA, USA) (#4695)	1:1000	
Anti-rabbit phospho-p44/42 MAPK Erk1/2 Thr202/Tyr204 D13.14.4E	Cell Signaling (Danvers, MA, USA) (#4370)	1:1000	
Anti-mouse Akt (pan) 40D4	Cell Signaling (Danvers, MA, USA) (#2920)	1:1000	
Anti-rabbit phospho-Akt (Ser473) D9E	Cell Signaling (Danvers, MA, USA) (#4060)	1:1000	
Anti-mouse phospho-histone H2A.X (Ser139)	Millipore (Burlington, MA, USA) (05-636-I)	1:500	
Anti-MYH7 (Myosin Heavy Chain 7)	Thermofisher (Waltham, MA, USA) (PA5-69132)		1:200
MF20 (Mouse Monoclonal Anti-Myosin Heavy Chain Antibody)	DSHB Hybridoma (Iowa City, Iowa, USA)		1:50
Rhodamine phalloidin	Thermofisher (Waltham, MA, USA) (R415)		1:40
HRP-labelled anti-mouse secondary antibody	GE Healthcare (Chicago, Illinois, USA) (NA931-1ML)	1:5000	
HRP-labelled anti-rabbit secondary antibody	GE Healthcare (Chicago, Illinois, USA) (NA934-1ML)	1:5000	
Goat anti-mouse Alexa Fluor 488	Thermofisher Scientific (Waltham, MA, USA) (A32723)		1:500
Goat anti-rabbit Alexa Fluor 594	Thermofisher Scientific (Waltham, MA, USA) (A32740)		1:500
Goat anti-rabbit Alexa Fluor 488	Thermofisher Scientific (Waltham, MA, USA) (A32731)		1:500
Goat anti-rabbit Alexa Fluor 488 (3D models)	Thermofisher (Waltham, MA, USA) (A11034)		1:200
**CELL CULTURE MEDIA**		**REAGENTS and PLASMIDS**	
C2C12 myoblasts/SIGMA (St. Louis, MI, USA) (91031101)		pX459 vector (pSpCas9(BB)-2A-Puro)/Addgene (Watertown, MA, USA) (#62988)	
DMEM (Dulbecco’s modified Eagle’s medium) high glucose/Invitrogen (Waltham, MA, USA) (61965-026)		Lipofectamine 3000/Invitrogen (Waltham, MA, USA) (L3000015)	
FBS (fetal bovine serum)/Sigma-Aldrich (St. Louis, MI, USA) (#F7524-500ML)		Puromycin/InvivoGen (San Diego, CA, USA) (ant-pr-1)	
Penicillin/streptomycin/Lonza (Basel, Switzerland) (#DE17-602E)		DNA polymerase/NZYTech (Lisbon, Portugal) (MB354)	
Medium 199/Invitrogen (Waltham, MA, USA) (41150020)		MiSeq DNA/Illumina (San Diego, CA, USA) (MS-102-2003)	
Fetuin/Life Technologies (Waltham, MA, USA) (10344026)		BCA system/Pierce (Waltham, MA, USA) (23227)	
hEGF/Life Technologies (Waltham, MA, USA) (PHG0311)		ECL western blotting system/Thermo Fisher Scientific (Waltham, MA, USA) (Pierce 32106)	
bFGF/Life Technologies (Waltham, MA, USA) (PHG0026)		Methanol/Panreac AppliChem (Barcelona, Spain) (#131091.1612)	
Insuline/Sigma (St. Louis, MI, USA) (91077C-1G)		BSA (bovine serum albumin)/Sigma-Aldrich (St. Louis, MI, USA) (#A7906)	
Dexamethasone/Sigma (St. Louis, MI, USA) (D4902-100mg)		PBS (phosphate-buffered saline)/Lonza (Basel, Switzerland) (#BE17-515Q)	
Horse serum/Thermofisher (Waltham, MA, USA) (#26050-088)		Goat serum/Sigma-Aldrich (St. Louis, MI, USA) (#G9023-10ML)	
**SOFTWARE AND PLATFORMS**		Donkey serum/Sigma-Aldrich (St. Louis, MI, USA) (#D9663-10ML)	
CRISPResso (http://crispresso.pinellolab.partners.org/)		Triton X-100/Sigma-Aldrich (St. Louis, MI, USA) (#X100-1L)	
TIDE (https://tide.deskgen.com/)		Prolong Gold with 4′,6-Diamidino-2-Phenylindole (DAPI)/Cell Signalling Technology (Danvers, MA, USA) (P36935)	
ImageJ (U.S. National Institutes of Health, Bethesda, Maryland, USA)		TBS (Tris-buffered saline)/Canvax Biotech (Cordoba, Spain) (BR0042)	
Prism 8 (GraphPad Software, Inc)		Hoechst 33324/Thermo Fisher Scientific (Waltham, MA, USA) (H3570)	

**Table 2 cells-09-01286-t002:** DNA sequences used in this work.

Name and 5′ to 3′ Sequence
sg10: CCGGCTGGCAGATGCCCTGCAGG
sg11: GCGGGCTCAGCATGAGGACCAGG
sg12: GGTCCACCTTGTCATATTCTTCC
ssODNmex4g10: GTGGAGATCGATAACGGGAAGCAGCGAGAGTTTGAGAGCCGGCTGGCAGATGCCCTGCAGGAGCTCTGGGCTCAGCATGAGGACCAGGTGGAACAGTATAAGAAGGAGCT
ssODNmex4g11: GAGTTTGAGAGCCGGCTGGCAGATGCCCTGCAGGAGCTCTGGGCTCAGCATGAGGACCAGGTGGAACAGTATAAGAAGGAGCTAGAAAAGACATACTCCGCCAAGGTGCT
ssODNmex4g12: AGAGCCGGCTGGCAGATGCCCTGCAGGAGCTCTGGGCTCAGCATGAGGACCAGGTGGAACAGTATAAGAAGGAGCTAGAAAAGACATACTCCGCCAAGTGCTGGCCTCAT
ssODNmex4g10mut: GTGGAGATCGATAACGGGAAGCAGCGAGAGTTTGAGAGCCGGCTGGCAGATGCCCTGCAAGAGCTCTGGGCTCAGCATGAGGACCAGGTGGAACAGTATAAGAAGGAGCT
ssODNmex4g10mut2: GTGGAGATCGATAACGGGAAGCAGCGAGAGTTTGAGAGCCGTCTTGCCGACGCACTTCAAGAGCTCTGGGCTCAGCATGAGGACCAGGTGGAACAGTATAAGAAGGAGCT
ssODNmex4g11mut: GAGTTTGAGAGCCGGCTGGCAGATGCCCTGCAGGAGCTCTGGGCTCAGCATGAGGACCAAGTGGAACAGTATAAGAAGGAGCTAGAAAAGACATACTCCGCCAAGGTGCT
ssODNmex4g11mut2: GAGTTTGAGAGCCGGCTGGCAGATGCCCTGCAGGAGCTCTGGGCACAACACGAAGATCAAGTGGAACAGTATAAGAAGGAGCTAGAAAAGACATACTCCGCCAAGGTGCT
ssODNmex4g12mut: AGAGCCGGCTGGCAGATGCCCTGCAGGAGCTCTGGGCTCAGCATGAGGATCAGGTGGAACAGTATAAGAAGGAGCTAGAAAAGACATACTCCGCCAAGTGCTGGCCTCAT
ssODNmex4g12mut2: AGAGCCGGCTGGCAGATGCCCTGCAGGAGCTCTGGGCTCAGCATGAGGATCATGTTGAGCAATACAAAAAAGAGCTAGAAAAGACATACTCCGCCAAGTGCTGGCCTCAT
DeepSeq-Fw: TCGTCGGCAGCGTCAGATGTGTATAAGAGACAGAGGCGAGTGGATGCTGAG
DeepSeq-Rv: GTCTCGTGGGCTCGGAGATGTGTATAAGAGACAGGTCAATGCGGATTCGAGACT
Sanger-mLmna_Ex4_Fw: CCAGGCTAAGCGAGGGCTGC
Sanger-mLmna_Ex4_Rv: CCTGAGGAAGGCATCCCTGG

**Table 3 cells-09-01286-t003:** Molecular properties of the selected clones.

ID	Type	CDS ^(1)^	Protein (Expected)
459-A6	Control	wt/wt	p.(666*)/p.(666*) (665 aa)
459-B4		wt/wt	p.(666*)/p.(666*) (665 aa)
g10L-A2	Null	c. 734_735del	p.(252*) (251 aa)
g11Hm-8		c. [757_758ins] [758_810del]	p.(263*) (262 aa)
g12Hm2-3		c. [767del] [811subsC > G]	p.(263*) (262 aa)
g12Lm2-3		c. [739_740ins] [740_761del] [766_767GT > AG]	p.(273*) (272 aa)
g12Lm2-5		c. [del767]/c. [del769]	p.(263*) (262 aa)/p.(263*) (262 aa)
g10L-A1	Delta1	c.734_736del (loss of 3 nt)	p.Leu245del (loss of 1 aa)
g11Hm2-1	Delta7-1	c.754_774del (loss of 21 nt)	p.His252_Gln258del (loss of 7 aa)
g11Hm2-10	Delta7-10	c.754_774del (loss of 21 nt)	p.His252_Gln258del (loss of 7 aa)
g10Hm-5	Delta14	c.707_748del (loss of 42 nt)	p.Glu236_Arg249del (loss of 14 aa)
g12Hm-6	Delta15	c.766_810del (loss of 45 nt)	p.Val256_Lys270del (loss of 15 aa)
g10H-A4	Delta57	c. [726_727ins] [727_1967del]/c. 640_810del	p.(252*) (251 aa)/p.Glu214_Lys270del (loss of 57 aa)
g11H-D6	R249W	c. [744_745subsGC > CT] [750subsT > G]	p.Arg249Tryp (665 aa)

^1^ Obtained by deep sequencing.

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
