# Peer review of "Consequences of Lmna Exon 4 Mutations in Myoblast Function"

_cells, 2020, doi:10.3390/cells9051286_

Round 1

Reviewer 1 Report

Review

 Consequences of Lmna exon 4 mutations in myoblast function

In their paper, Gomez-Dominguez and co-workers report the characterization of a collection of Lmnaexon 4 mutants. The mutants were obtained using CRISPR/Cas9technology in mouse C2C12 myoblasts. The authors show abnormal nuclei shape, altered distribution of several components of the nuclear envelope and the LINC complex and decreased myogenic differentiation capacity in mutant myoblasts compared to control cells.

The methods are sound, the cellular models represent a valuable tool to study mutation consequences, the results are interesting as they provide insights in mutation consequences that could be useful for gene therapy. The paper is globally concise and well written. However, I have several concerns which should be addressed before publication.

Major points

Despite the authors made considerable effort to characterize their models, we think that the first part of the paper reporting lamins and emerin distribution and nuclear morphology should be shortened as it is mostly confirmatory. In contrast, the experiments dealing with signaling pathways should be more developed and consolidate. More precisely, the results concerning ERK1 and AKT signaling pathways are too heterogeneous to be shown or to draw concrete conclusions. The experiments should be repeated to improve repeatability or performed again with another method like qPCR. If not possible to strengthen, the results of figure 7 should be removed.

In addition, we would expect to find results about other signaling pathways, at least about the WNT/b-catenin signaling pathway which has been also associated with Lamin A mutants in the literature as well as the MAPK signaling pathway. We suggest testing at least the beta-catenin expression.

We wonder if the authors have checked that the myoblasts they use for experiments, are not partially differentiated using, for example, MHC staining which should remain negative (or any other maker for differentiation). This would be useful to confirm that differences between controls and mutants cell lines are not due to differentiation. It could also explain the large heterogeneity observed in experiments addressing signaling pathways.

Other points

Introduction:

-Lamins structure is presented briefly but nothing is mentioned about their functions whereas some hypothesis are evoked regarding the mechanism for lamin mutant effect on cellular functions (line56). A brief description of lamin functions should be added.

-Line 65 : HGPS is mentioned as an example for gene therapy research but this laminopathy has not been introduced in the text. Although it is the most famous type of laminopathy, the pathophysiological mechanism of the disease should be briefly reminded.

-The authors should briefly explain why they choose the R249W mutation among the 47 mutations described in exon 4. I understand that it is mainly because it produces a severe congenital muscular dystrophy but is it the only one? Is there any paper reporting this mutation previously with effects characterization? It should be indicated and if yes, indicate also in the results section if the features observed are concordant with the literature.

Materials and methods

-This section should be more concise: as an example, create a section « chemicals and antibodies », this would avoid detail those products in other sections. The sections Immunofluorescence microscopy, Nuclear morphology analyses and Myogenic differentiation should be merged.

-regarding human cell lines, a reference is given (Bertrand, et al 2014). It is not clear for which purpose the reference is given? Is it a gift from the lab?

-Statistical analysis: the authors used a Student t-test. I suggest that they check if it is the right test or if a Mann Withney should be used instead.

Results:

-Table 2: A column should be added mentioning which mutants have been described in humans and what was the phenotype associated

-Figures 2,3 and 4 should be merged under a section which would be something like: the new models are reproducing usual features described in laminopathies

As a general comment for the legends to the figures: although the whole paper is dealing with myoblasts, it would be useful to mention in each legend, that the observed mutant cells are myoblasts.

-Figure 2:

-In panel B, the plots showing average Lamin A/C fluorescence intensity across nuclei by mutant type could be merged since there is a different color for each mutant type (this stands for the other plots in other figure).

-In addition, the zoom showed in panel C could be shown on a picture of panel B (no need to show new pictures)

Results section 3 :

-Did the authors check for lamin B1 and Emerin expression level like they did for Lamin A? From what I can see on the IF pictures, it seems that expression levels are lower for the mutants. The authors should address this point

Results section 5 :

The authors chose to describe the differentiation capacity at day 5. What happened after day 5? Before? Is the differentiation delayed or inhibited?

The accent should be put on the effect of Delta-exon 4 mutant which preserves myogenic differentiation.

Figure 5b is not clear to me: first the controls are referred with numbers (whereas they are referred as “control” in all other figures), second I don’t see the green staining specific for myogenic differentiation in controls whereas I can see it in pictures corresponding to the mutant R249W. This is not in accordance with the text.

Section 8:

The authors should perform IF with specific gH2AX antibodies to check that the extraction buffer they used for proteins extraction is adapted to extract chromatin linked proteins. Did they perform these experiments at different passages or differentiation steps? Has this feature been previously described for other missense mutations leading to abnormal protein with a gain of function?

-Discussion:

-Line 418 : « We have no clear explanation for this apparent contradiction and, therefore, further studies are needed to solve it. » Some suggestions could be made from the literature

-A paper from Janin et al reported recently myoblasts premature differentiation induced by Lmna∆8-11 . It should be discussed in the discussion section.

-the authors should discuss the various consequences of missense mutations which could lead to lamin or prelamin accumulation compared to null mutations

Minor points:

-Almost 500 mutations in the introduction and more than 400 in the abstract: although it is not contradictory, it should be more similar.

-On figure 3B, scale bar is missing in the legend

-Line 94 : FBS = fœtal bovin serum. No need to mention it again line 100.

-Fig 5a: the legend for the tested mutants should be completed for one of the “delta” (the reference number is missing).

Reviewer 2 Report

This manuscript “Consequences of Lmna exon 4 mutations in myoblast function” describes the generation and analysis on C2C12 clones that have been genome edited to either lack expression of LMNA, or to be mutated with multiple deletion sizes or a missense mutation, all impacting exon 4. These cells are characterized in various ways to explore the impact of the mutations on known roles of LMNA in these cells. Overall, the findings are largely expected, although sometimes unexpected as in the case of p-ERK/p-AKT levels. The value of these predominantly artificial mutations in comparison to actual human mutations in this exon is unclear, although there is relevance of the exon skipping as a therapeutic strategy. The results are often clearly presented, although there are some unexpected findings that question the approach/methods and need addressing.

Specific comments:

Are the CRISPR-mediated mutations impacting one allele or all alleles? If multiple, how would the random mutagenesis from an indel be identical for all three alleles. How would this compare to the predominantly autosomal dominant etiology of LMNA-associated laminopathies of the striated muscle.

Why do the IF and WB levels of the lamin A/C not match up? There is a lot of lamin in the WB of controls and almost none in the mutants. But the difference is not nearly as much in the IFs. Were different exposures used?

For the p-ERK/p-AKT levels experiments, could the authors include a control where they knock down LMNA by siRNA to see if the lack of activation is due to differences in the mutations used in this study compared to approaches used elsewhere, or if the increased levels simply are not reproducible, at least under the conditions used in the study. As for the cell type explanation, they don’t seem to hold up-see PMID:19022376

It is confusing why there are no differences in the WB for Sun1/Sun2, but by IF there is a considerable decrease in the protein detection. Also, I recall that Sun2, but certainly not Sun1, was at best variably LMNA dependent in its NE localization. Sun1’s NE localization is LMNA independent according to numerous reports.

Reviewer 3 Report

The manuscript submitted by Gomez-Dominguez et al. entitled, “Consequences of Lmna exon 4 mutations in myoblast function” describes a possible molecular cause underlying laminopathic pleiotropy. In their well-focused study, the authors compare several myoblast clones genome edited to harbor different types of exon 4 deletions. Lamin expression/distribution, nuclear morphology, and sarcomerogenesis/myogenic differentiation were evaluated in both two and three dimensional cultures. In addition, normal myoblasts and a myoblast line edited to carry the pathological R249W variant was included as part of the present study. The authors show evidence that null mutants, indels and the R249W variant exhibited dysregulated expression and localization of lamin and other nuclear envelope components including emerin and SUN1. In particular, the unique DNA damage phenotype associated with the R249W mutation is a novel observation that establishes the foundation for understanding the possible mechanism underlying laminopathy associated with R249W.

I have some questions/comments for the authors to address:

  • Pg. 3, line 96 – the authors describe two controls isolated from individuals of 25 and 88 years of age; in addition, the third line harboring the R249W variant was isolated from a 3 year old patient. In light of this:
    • What age related differences are observed at baseline?
    • What epigenomic modifications, if any, are different among the 3, 25, and 88 year old samples?
  • Pg. 11, line 331 – What is the explanation for what seems to be compensatory sarcomerogenesis in the Delta57 clone?
  • Pg. 12, line 371 – Is the high level of DNA damage due to increased damage or decreased repair?
  • Are the changes in the nuclear lamina and the LINC complex accompanied by any changes in the NPC? As all three are major integrated components of the nuclear envelope, and it is known that the nuclear lamina and nuclear pore complexes contribute to DSB repair, do disruptions to the nuclear lamina affect the structure/function of the NPC that affects its ability to mediate DNA repair?
  • With the increased DNA damage associated with the R249W variant, are there any reports of uncontrolled proliferation/tumorigenesis associated with cells that harbor this mutation?
  • Figure 8 – please correct the figure legend to read as γH2AX and α-tubulin as it does in the figure. Presently it is written as ©-H2AX and ©-tubulin

Minor comments:

  • Syntax/style errors throughout the text, please correct, e.g.
    • Pg. 2, line 71 – should read “…linker domain comprised of the two central…” instead of “comprised between…”
    • Pg.2, line 91 – “the muscle of a tip on a C3H mouse line.” These are myoblasts originally established from satellite cells derived from the thigh muscle of a female C3H murine donor following a crush injury (Yaffe and Saxel, 1977). This is not clear from the way it is currently written in the text.

Round 2

Reviewer 2 Report

The authors have addressed my prior concerns in this revision.

Reviewer 3 Report

All concerns have been satisfactorily addressed.